# Learning Task-General Representations with Generative Neuro-Symbolic Modeling

**Reuben Feinman & Brenden M. Lake**
New York University
{reuben.feinman,brenden}@nyu.edu

## Abstract

People can learn rich, general-purpose conceptual representations from only raw perceptual inputs. Current machine learning approaches fall well short of these human standards, although different modeling traditions often have complementary strengths. Symbolic models can capture the compositional and causal knowledge that enables flexible generalization, but they struggle to learn from raw inputs, relying on strong abstractions and simplifying assumptions. Neural network models can learn directly from raw data, but they struggle to capture compositional and causal structure and typically must retrain to tackle new tasks. We bring together these two traditions to learn generative models of concepts that capture rich compositional and causal structure, while learning from raw data. We develop a generative neuro-symbolic (GNS) model of handwritten character concepts that uses the control flow of a probabilistic program, coupled with symbolic stroke primitives and a symbolic image renderer, to represent the causal and compositional processes by which characters are formed. The distributions of parts (strokes), and correlations between parts, are modeled with neural network subroutines, allowing the model to learn directly from raw data and express nonparametric statistical relationships. We apply our model to the Omniglot challenge of human-level concept learning, using a background set of alphabets to learn an expressive prior distribution over character drawings. In a subsequent evaluation, our GNS model uses probabilistic inference to learn rich conceptual representations from a single training image that generalize to 4 unique tasks, succeeding where previous work has fallen short.

## 1 Introduction

Human conceptual knowledge supports many capabilities spanning perception, production and reasoning [37]. A signature of this knowledge is its productivity and generality: the internal models and representations that people develop can be applied flexibly to new tasks with little or no training experience [30]. Another distinctive characteristic of human conceptual knowledge is the way that it interacts with raw signals: people learn new concepts directly from raw, high-dimensional sensory data, and they identify instances of known concepts embedded in similarly complex stimuli. A central challenge is developing machines with these human-like conceptual capabilities.

Engineering efforts have embraced two distinct paradigms: *symbolic* models for capturing structured knowledge, and *neural network* models for capturing nonparametric statistical relationships. Symbolic models are well-suited for representing the causal and compositional processes behind perceptual observations, providing explanations akin to people's intuitive theories [38]. Quintessential examples include accounts of concept learning as program induction [13, 46, 29, 15, 4, 28]. Symbolic programs provide a language for expressing causal and compositional structure, while probabilistic modeling offers a means of learning programs and expressing additional conceptual knowledge through priors. The Bayesian Program Learning (BPL) framework [29], for example, provides a dictionary of simple sub-part primitives for generating handwritten character concepts, and symbolic relations that specify how to combine sub-parts into parts (strokes) and parts into whole character concepts. These abstractions support inductive reasoning and flexible generalization to a range of different tasks, utilizing a single conceptual representation [29].

BPL model
(centered)  GNS model  Humans

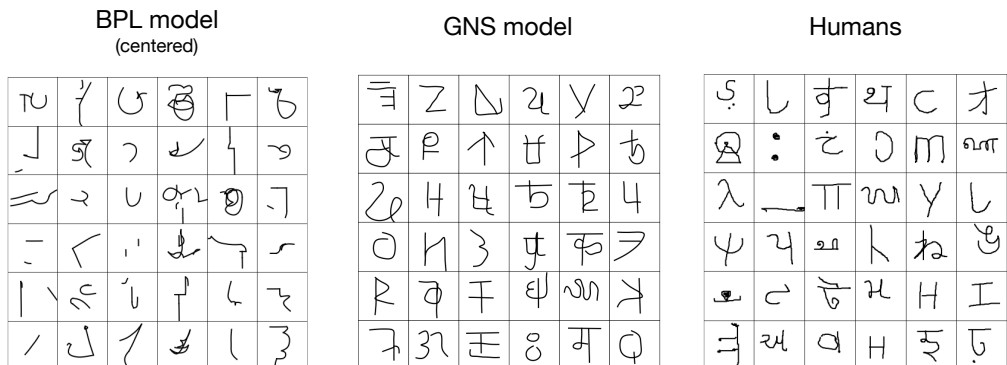

Figure 1: Character drawings produced by the BPL model (left), GNS model (middle), and humans (right).

Symbolic models offer many useful features, but they come with important limitations. Foremost, symbolic probabilistic models make simplifying and rigid parametric assumptions, and when the assumptions are wrong—as is common in complex, high-dimensional data—they create bias [11]. The BPL character model, for example, assumes that parts are largely independent a priori, an assumption that is not reflective of real human-drawn characters. As a consequence, characters generated from the raw BPL prior lack the complexity of real characters (Fig 1, left), even though the posterior samples can appear much more structured. Another limitation of symbolic probabilistic models is that the construction of structured hypothesis spaces requires significant domain knowledge [2]. Humans, meanwhile, build rich internal models directly from raw data, forming hypotheses about the conceptual features and the generative syntax of a domain. As one potential resolution, previous work has demonstrated that the selection of structured hypotheses can itself be attributed to learning in a Bayesian framework [47, 13, 14, 41, 24, 40]. Although more flexible than a priori structural decisions, models of this kind still make many assumptions, and they have not yet tackled the types of raw, high-dimensional stimuli that are distinctive of the neural network approach.

The second paradigm, neural network modeling, prioritizes powerful nonparametric statistical learning over structured representations. This modeling tradition emphasizes *emergence*, the idea that conceptual knowledge arises from interactions of distributed sub-symbolic processes [36, 32]. Neural networks are adept at learning from raw data and capturing complex patterns. However, they can struggle to learn the compositional and causal structure in how concepts are formed [30]; even when this structure is salient in the data, they may have no obvious means of incorporating it. These limitations have been linked to shortcomings in systematic generalization [35, 27] and creative abilities [31]. An illustrative example is the *Omniglot challenge*: in 4 years of active research, neural network models do not yet explain how people quickly grasp new concepts and use them in a variety of ways, even with relatively simple handwritten characters [31]. Surveying over 10 neural models applied to Omniglot, Lake et al. [31] found that only two attempted both classification and generation tasks, and they were each outperformed by the fully-symbolic, probabilistic BPL. Moreover, neural generative models tended to produce characters with anomalous characteristics, highlighting their shortcomings in modeling causal and compositional structure (see Fig. A13 and [31, Fig. 2a]).

In this paper, we introduce a new approach that leverages the strengths of both the symbolic and neural network paradigms by representing concepts as probabilistic programs with neural network subroutines. We describe an instance of this approach developed for the *Omniglot challenge* [29] of task-general representation learning and discuss how we see our Omniglot model fitting into a broader class of Generative Neuro-Symbolic (GNS) models that seek to capture the data-generation process. As with traditional probabilistic programs, the control flow of a GNS program is an explicit representation of the *causal* generative process that produces new concepts and new exemplars. Moreover, explicit re-use of parts through repeated calls to procedures such as `GeneratePart` (Fig. 2) ensures a representation that is *compositional*, providing an appropriate inductive bias for compositional generalization. Unlike fully-symbolic probabilistic programs, however, the distribution of parts and correlations between parts in GNS are modeled with neural networks. This architectural choice allows the model to learn directly from raw data, capturing nonparametric statistics while requiring only minimal prior knowledge.

Table 1: Attempted Omniglot tasks by model. Attempt does not imply successful completion.

| Task | BPL [29] | RCN [12] | VHE [21] | SG [43] | SPIRAL [8] | Matching Net [49] | MAML [7] | Graph Net [9] | Prototypical Net [45] | ARC [44] |
|---|---|---|---|---|---|---|---|---|---|---|
| One-shot classification | x | x | x | | | x | x | x | x | x |
| Parsing | x | | | | x | | | | | |
| Generate exemplars | x | x | x | x | | | | | | |
| Generate concepts (type) | x | | x | x | | | | | | |
| Generate concepts | x | | | x | x | | | | | |

We develop a GNS model for the Omniglot challenge of learning flexible, task-general representations of handwritten characters. We report results on 4 Omniglot challenge tasks with a single model: 1) one-shot classification, 2) parsing/segmentation, 3) generating new exemplars, and 4) generating new concepts (without constraints); the 5th and final task of generating new concepts (from type) is left for future work. We also provide log-likelihood evaluations of the generative model. Notably, our goal is not to chase state-of-the-art performance on one task across many datasets (e.g., classification). Instead we build a model that learns deep, task-general knowledge within a single domain and evaluate it on a range of different tasks. This "deep expertise" is arguably just as important as "broad expertise" in characterizing human-level concept learning [37, 31]; machines that seek human-like abilities will need both. Our work here is one proposal for how neurally-grounded approaches can move beyond pattern recognition toward the more flexible model-building abilities needed for deep expertise [30]. In Appendix E, we discuss how to extend GNS to another conceptual domain, providing a proposal for a GNS model of 3D object concepts.

## 2 RELATED WORK

The Omniglot dataset and challenge has been widely adopted in machine learning, with models such as Matching Nets [49], MAML [7], and ARC [44] applied to just one-shot classification, and others such as DRAW [19], SPIRAL [8], and VHE [21] applied to one or more generative tasks. In their "3-year progress report," Lake et al. [31] reviewed the current progress on Omniglot, finding that although there was considerable progress in one-shot classification, there had been little emphasis placed on developing task-general models to match the flexibility of human learners (Table 1). Moreover, neurally-grounded models that attempt more creative generation tasks were shown to produce characters that either closely mimicked the training examples or that exhibited anomalous variations, making for easy identification from humans (see Fig. A13 and [31, Fig. 2a]). Our goal is distinct in that we aim to learn a single neuro-symbolic generative model that can perform a variety of unique tasks, and that generates novel yet structured new characters.

Neuro-symbolic modeling has become an active area of research, with applications to learning input-output programs [42, 17, 3, 39, 48], question answering [50, 34] and image description [26, 4]. GNS modeling distinguishes itself from prior work through its focus on hybrid generative modeling, combining both structured program execution and neural networks directly in the probabilistic generative process. Neuro-symbolic VQA models [50, 34] are neither generative nor task-general; they are trained discriminatively to answer questions. Other neuro-symbolic systems use neural networks to help perform inference in a fully-symbolic generative model [26, 4], or to parameterize a prior over fully-symbolic hypotheses [22]. In order to capture the dual structural and statistical characteristics of human conceptual representations, we find it important to include neural nets directly in the forward generative model. As applied to Omniglot, our model bears some resemblance to SPIRAL [8]; however, SPIRAL does not provide a density function, and it has no hierarchical structure, limiting its applications to image reconstruction and unconditional generation.

Another class of models on the neuro-symbolic spectrum aims to learn "object representations" with neural networks [5, 18, 25], which add minimal object-like symbols to support systematic reasoning and generalization. Although these models have demonstrated promising results in applications such as scene segmentation and unconditional generation, they have not yet demonstrated the type of rich inductive capabilities that we are after: namely, the ability to learn "deep" conceptual representations from just one or a few examples that support a variety of discriminative and generative tasks.

Other works (e.g. [16, 20]) have used autoregressive models like ours with similar stroke primitives to model the causal generative processes of handwriting. We develop a novel architecture for generating

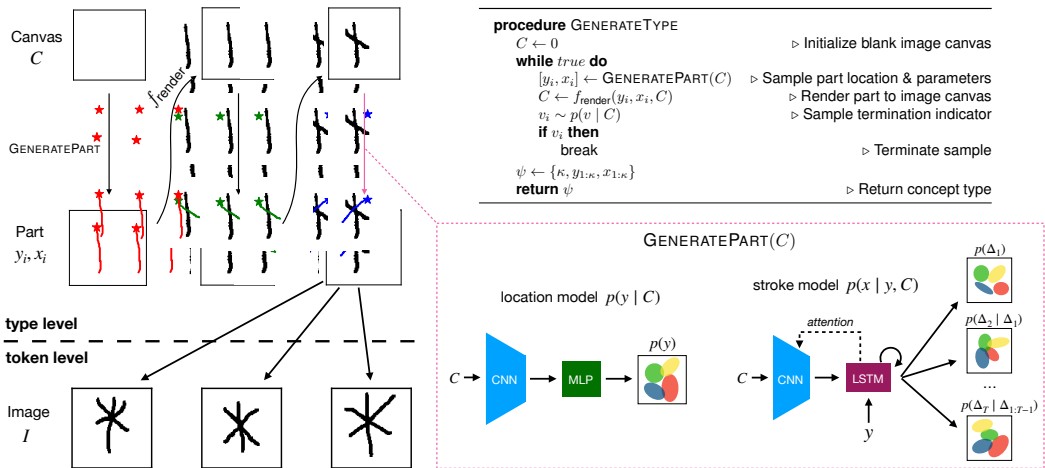

Figure 2: A generative neuro-symbolic (GNS) model of character concepts. The type model `GenerateType` ($P(\psi)$) produces character types one stroke at a time, using an image canvas $C$ as memory. At each step, the current canvas $C$ is fed to procedure `GeneratePart` and a stroke sample is produced. The canvas is first processed by the *location model*, a CNN-MLP architecture that samples starting location $y$, and next by the *stroke model*, a CNN-LSTM architecture that samples trajectory $x$ while attending to the encoded canvas. Finally, a symbolic renderer updates the canvas according to $x$ and $y$, and a *termination model* decides whether to terminate the type sample. Unique exemplars are produced from a character type by sampling from the token model conditioned on $\psi$, adding motor noise to the drawing parameters and performing a random affine transformation.

handwriting, which represents explicit compositional structure by modeling parts and relations with separate modules and applying intermediate symbolic rendering. Most importantly, these prior models have not made a connection to the image; therefore while they can generate handwriting as symbolic coordinates, they cannot explain how people use their causal knowledge to learn new characters from visual presentations, how they infer the strokes of a character seen on paper, or how they generate a new example of an observed character. By combining a powerful autoregressive model of handwriting with a symbolic image renderer and algorithms for probabilistic inference, we seek to replicate a spectrum of unique human concept learning abilities.

## 3 GENERATIVE MODEL

Our GNS model leverages the type-token hierarchy of BPL [29], which offers a useful scaffolding for conceptual models (Fig. 2, left).[1] The type-level model $P(\psi)$ defines a prior distribution over character concepts, capturing overarching principles and regularities that tie together characters from different alphabets and providing a procedure to generate new character concepts unconditionally in latent stroke format (Fig. 2, right). A token-level model $p(\theta|\psi)$ captures the within-class variability that arises from motor noise and drawing styles, and an image distribution $P(I|\theta)$ provides an explicit model of how causal stroke actions translate to image pixels. All parameters of our model are learned from the Omniglot background set of drawings (Appendix A). The full joint distribution over type $\psi$, token $\theta^{(m)}$ and image $I^{(m)}$ factors as

$$P(\psi, \theta^{(m)}, I^{(m)}) = P(\psi)P(\theta^{(m)}|\psi)P(I^{(m)}|\theta^{(m)}). \tag{1}$$

Although sharing a common hierarchy, the implementation details of each level in our GNS model differ from BPL in critical ways. The GNS type prior $P(\psi)$ is a highly expressive generative model that uses an external image canvas, coupled with a symbolic rendering engine and an attentive

---

[1]Aspects of our model were recently published in a non-archival conference proceedings [6]. The previous manuscript presents only a prior distribution that alone performs just one task (generating new concepts); our new developments include a full hierarchical model, a differentiable image renderer / likelihood, and a procedure for approximate probabilistic inference from image data. These ingredients together enable GNS to perform 4 unique conceptual tasks.

recurrent neural network, to condition future parts on the previous parts and model sophisticated causal and correlational structure. This structure is essential to generating new character concepts in realistic, human-like ways (Sec. 5). Moreover, whereas the BPL model is provided symbolic relations for strokes such as "attach start" and "attach along," GNS learns implicit relational structure from the data, identifying salient patterns in the co-occurrences of parts and locations. Importantly, the GNS generative model is designed to be differentiable at all levels, yielding log-likelihood gradients that enable powerful new inference algorithms (Sec. 4) and estimates of marginal image likelihood (Sec. 5).

**Type prior.** The type prior $P(\psi)$ is captured by a neuro-symbolic generative model of character drawings. The model represents a character as a sequence of strokes (parts), with each stroke $i$ decomposed into a starting location $y_i \in \mathbb{R}^2$ and a variable-length trajectory $x_i \in \mathbb{R}^{d_i \times 2}$. Rather than use raw pen trajectories as our stroke format, we use a *minimal spline* representation of strokes, obtained from raw trajectories by fitting cubic b-splines with a residual threshold. The starting location $y_i$ therefore conveys the first spline control point, and trajectory $x_i = \{\Delta_{i1}, ..., \Delta_{id_i}\}$ conveys the offsets between subsequent points of a $(d_i+1)$-length spline. These offsets are transformed into a sequence of relative points $x_i = \{x_{i1}, ..., x_{id_i+1}\}$, with $x_{i1} = 0$, specifying locations relative to $y_i$.

The model samples a type one stroke at a time, using an image canvas $C$ as memory to convey the sample state. At each step, a starting location for the next stroke is first sampled from the *location model*, followed by a trajectory from the *stroke model*. The stroke is then rendered to the canvas $C$, and a *termination model* decides whether to terminate or continue the sample. Each of the three model components is expressed by a neural network, using a LSTM as the stroke model to generate trajectories as in [16]. The details of these neural modules are provided in Appendix A. The type model $P(\psi)$ specifies an auto-regressive density function that can evaluate exact likelihoods of character drawings, and its hyperparameters (the three neural networks) are learned from the Omniglot background set of 30 alphabets using a maximum likelihood objective. A full character type $\psi$ includes the random variables $\psi = \{\kappa, y_{1:\kappa}, x_{1:\kappa}\}$, where $\kappa \in \mathbb{Z}^+$ is the number of strokes. The density function $P(\psi)$ is also fully differentiable w.r.t. the continuous random variables in $\psi$.

**Token model.** A character token $\theta^{(m)} = \{y_{1:\kappa}^{(m)}, x_{1:\kappa}^{(m)}, A^{(m)}\}$ represents a unique instance of a character concept, where $y_{1:\kappa}^{(m)}$ are the token-level locations, $x_{1:\kappa}^{(m)}$ the token-level parts, and $A^{(m)} \in \mathbb{R}^4$ the parameters of an affine warp transformation. The token distribution factorizes as

$$P(\theta^{(m)}|\psi) = P(A^{(m)}) \prod_{i=1}^{\kappa} P(y_i^{(m)} \mid y_i)P(x_i^{(m)} \mid x_i). \tag{2}$$

Here, $P(y_i^{(m)} \mid y_i)$ represents a simple noise distribution for the location of each stroke, and $P(x_i^{(m)} \mid x_i)$ for the stroke trajectory. The first two dimensions of affine warp $A^{(m)}$ control a global re-scaling of the token drawing, and the second two a global translation of its center of mass. The distributions and pseudocode of our token model are given in Appendix A.

**Image model.** The image model $P(I^{(m)} \mid \theta^{(m)})$ is based on [29] and is composed of two pieces. First, a differentiable symbolic engine $f$ receives the token $\theta^{(m)}$ and produces an image pixel probability map $p_{\text{img}} = f(\theta^{(m)}, \sigma, \epsilon)$ by evaluating each spline and rendering the stroke trajectories. Here, $\sigma \in \mathbb{R}^+$ is a parameter controlling the rendering blur around stroke coordinates, and $\epsilon \in (0, 1)$ controlling pixel noise, each sampled uniformly at random. The result then parameterizes an image distribution $P(I^{(m)} \mid \theta^{(m)}) = \text{Bernoulli}(p_{\text{img}})$, which is differentiable w.r.t. $\theta^{(m)}$, $\sigma$, and $\epsilon$.

## 4 PROBABILISTIC INFERENCE

Given an image $I$ of a novel concept, our GNS model aims to infer the latent causal, compositional process for generating new exemplars. We follow the high-level strategy of BPL for constructing a discrete approximation $Q(\psi, \theta \mid I)$ to the desired posterior distribution [29],

$$P(\psi, \theta \mid I) \approx Q(\psi, \theta \mid I) = \sum_{k=1}^{K} \pi_k \delta(\theta - \theta_k)\delta(\psi - \psi_k). \tag{3}$$

A heuristic search algorithm is used to find $K$ good parses, $\{\psi, \theta\}_{1:K}$, that explain the underlying image with high probability. These parses are weighted by their relative posterior probability,

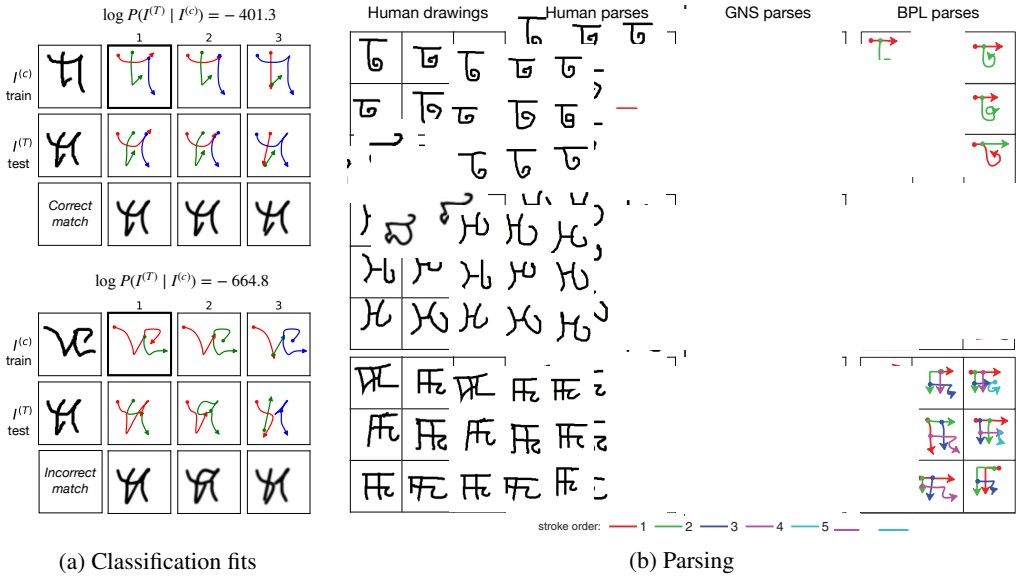

(a) Classification fits          (b) Parsing

Figure 3: Classification fits and parsing. (a) Posterior parses from two training images were refit to the same test image. The first row of each grid shows the training image and its top-3 predicted parses (best emboldened). The second row shows the test image and its re-fitted training parses. Reconstructed test images are shown in the final row. The correct training image reports a high forward score, indicating that $I^{(T)}$ is well-explained by the motor programs for this $I^{(c)}$. (b) 27 character images from 3 classes are shown alongside their ground truth human parses, predicted parses from the GNS model, and predicted parses from the BPL model.

$\pi_k \propto \tilde{\pi}_k = P(\psi_k, \theta_k, I)$ such that $\sum_k \pi_k = 1$. To find the $K$ good parses, search uses fast bottom-up methods to propose many variants of the discrete variables, filtering the most promising options, before optimizing the continuous variables with gradient descent (we use $K = 5$). See Appendix B for further details about inference, and Appendix C for generating new exemplars of a concept.

**Inference for one-shot classification.** In one-shot classification, models are given a single training image $I^{(c)}$ from each of $c = 1, ..., C$ classes, and asked to classify test images according to their corresponding training classes. For each test image $I^{(T)}$, we compute an approximation of the Bayesian score $\log P(I^{(T)} \mid I^{(c)})$ for every example $I^{(c)}$, using our posterior parses $\{\psi, \theta^{(c)}\}_{1:K}$ and corresponding weights $\pi_{1:K}$ from $I^{(c)}$ (Eq. 3). The approximation is formulated as

$$\log P(I^{(T)} \mid I^{(c)}) \approx \log \int P(I^{(T)}|\theta^{(T)})P(\theta^{(T)} \mid \psi)Q(\psi, \theta^{(c)}, \mid I^{(c)})\partial\psi\partial\theta^{(c)}\partial\theta^{(T)}$$

$$\approx \log \sum_{k=1}^{K} \pi_k \max_{\theta^{(T)}} P(I^{(T)} \mid \theta^{(T)})P(\theta^{(T)} \mid \psi_k), \tag{4}$$

where the maximum over $\theta^{(T)}$ is determined by refitting token-level parameters $\theta^{(c)}$ to image $I^{(T)}$ with gradient descent. Following Lake et al. [29], we use a two-way version of the Bayesian score that also considers parses of $I^{(T)}$ refit to $I^{(c)}$. The classification rule is therefore

$$c^* = \arg\max_c \log P(I^{(T)} \mid I^{(c)})^2 = \arg\max_c \log \Big[\frac{P(I^{(c)} \mid I^{(T)})}{P(I^{(c)})}P(I^{(T)} \mid I^{(c)})\Big], \tag{5}$$

where $P(I^{(c)}) \approx \sum_k \tilde{\pi}_k$ is approximated from the unnormalized weights of $I^{(c)}$ parses.

## 5   EXPERIMENTS

GNS was evaluated on four concept learning tasks from the Omniglot challenge [31]: one-shot classification, parsing, generating new exemplars, and generating new concepts. All evaluations use novel characters from completely held-out alphabets in the Omniglot evaluation set. As mentioned

earlier, our goal is to provide a single model that captures deep knowledge of a domain and performs strongly in a wide range of tasks, rather than besting all models on every task. Our experiments include a mixture of quantitative and qualitative evaluations, depending on the nature of the task.

**One-shot classification.** GNS was compared with alternative models on the one-shot classification task from Lake et al. [29]. The task involves a series of 20-way within-alphabet classification episodes, with each episode proceeding as follows. First, the machine is given one training example from each of 20 novel characters. Next, the machine must classify 20 novel test images, each corresponding to one of the training classes. With 20 episodes total, the task yields 400 trials. Importantly, all character classes in an episode come from the same alphabet as originally proposed [29], requiring finer discriminations than commonly used between-alphabet tests [31].

As illustrated in Fig. 3a, GNS classifies a test image by choosing the training class with the highest Bayesian score (Eq. 5). A summary of the results is shown in Table 2. GNS was compared with other machine learning models that have been evaluated on the within-alphabets classification task [31]. GNS achieved an overall test error rate of 5.7% across all 20 episodes (N=400). This result is very close to the original BPL model, which achieved 3.3% error with significantly more hand-design. The symbolic relations in BPL's token model provide rigid constraints that are key to its strong classification performance [29]. GNS achieves strong classification performance while emphasizing the nonparametric statistical knowledge needed for creative generation in subsequent tasks. Beyond BPL, our GNS

Table 2: Test error on within-alphabet one-shot classification.

| Model | Error |
| --- | --- |
| GNS | 5.7% |
| BPL [29] | 3.3% |
| RCN [12] | 7.3% |
| VHE [21] | 18.7% |
| Proto. Net [45] | 13.7% |
| ARC [44] | 1.5%* |

*used 4x training classes

model outperformed all other models that received the same background training. The ARC model [44] achieved an impressive 1.5% error, although it was trained with four-fold class augmentation and many other augmentations, and it can only perform this one task. In Appendix Fig. A10, we show a larger set of classification fits from GNS, including examples of misclassified trials.

**Parsing.** In the Omniglot parsing task, machines must segment a novel character into an ordered set of strokes. These predicted parses can be compared with human ground-truth parses for the same images. The approximate posterior of GNS yields a series of promising parses for a new character image, and to complete the parsing task, we identify the maximum a posteriori parse $k^* = \max_k \pi_k$, reporting the corresponding stroke configuration. Fig. 3b shows a visualization of the GNS predicted parses for 27 different raw images drawn from 3 unique character classes, plotted alongside ground-truth human parses (how the images were actually drawn) along with predicted parses from the BPL model. Compared to BPL, GNS parses possess a few unique desirable qualities. The first character class has an obvious segmentation to the human eye—evidenced by the consistency of human parses in all examples—and the GNS model replicates this consistency across all 9 predicted parses. In contrast, BPL predicts seemingly-unlikely parses for 2 of the examples shown. The second character is more complex, and it was drawn in different styles by different human subjects. The GNS model, which is trained on data from subjects with different styles, captures the uncertainty in this character by predicting a variety of unique parses. BPL, on the other hand, produces a single, ubiquitous segmentation across all 9 examples. In Appendix Fig. A11, we provide a larger set of parses from the GNS model for a diverse range of Omniglot characters.

**Generating new exemplars.** Given just one training image of a novel character concept, GNS produces new exemplars of the concept by sampling from the approximate conditional $P(I^{(2)}, \theta^{(2)} \mid I^{(1)})$ of Eq. 8. In Fig. 4a we show new exemplars produced by GNS for a handful of target images, plotted next to human productions (more examples in Appendix Fig. A12). In the majority of cases, samples from the model demonstrate that it has successfully captured the causal structure and invariance of the target class. In contrast, deep generative models applied to the same task miss meaningful compositional and causal structure, producing new examples that are easily discriminated from human productions [43, 21] (see Appendix Fig. A13). In some cases, such as the third column of Fig. 4a, samples from GNS exhibit sloppy stroke junctions and connections. Compared to BPL, which uses engineered symbolic relations to enforce rigid constraints at stroke junctions, GNS misses some of these structural elements. Nevertheless, new examples from GNS appear strong enough to pass for human in many cases, which we would like to test in future work with visual Turing tests.

---

Concept learning experiments can be reproduced using our pre-trained generative model and source code: `https://github.com/rfeinman/GNS-Modeling`.

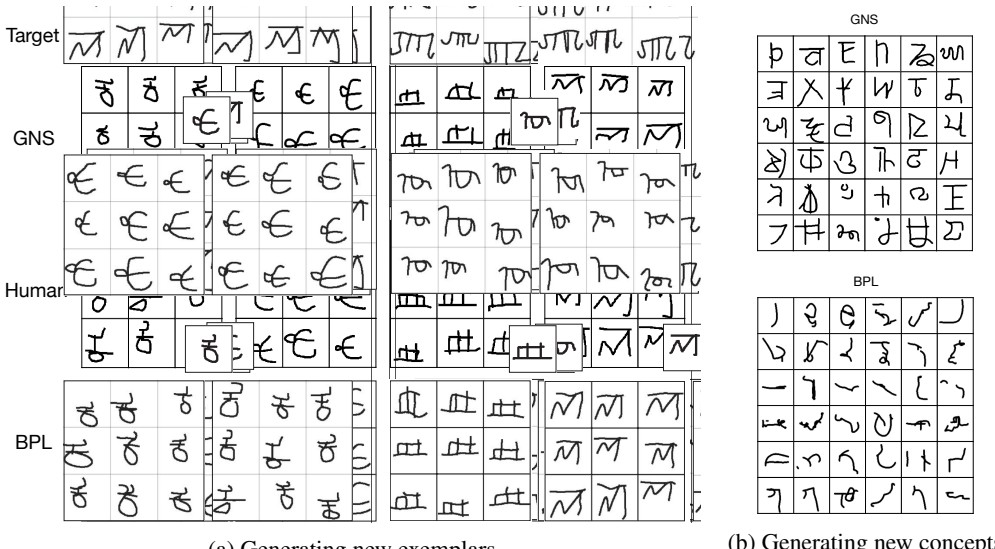

(a) Generating new exemplars      (b) Generating new concepts

Figure 4: Generation tasks. (a) GNS produced 9 new exemplars for each of 5 target images, plotted here next to human and BPL productions. (b) A grid of 36 new character concepts sampled unconditionally from GNS, shown next to BPL samples.

**Generating new concepts (unconstrained).** In addition to generating new exemplars of a particular concept, GNS can generate new character concepts altogether, unconditioned on training images. Whereas the BPL model uses a complicated procedure for unconditional generation that involves a preliminary inference step and a supplemental nonparametric model, GNS generates new concepts by sampling directly from the type prior $P(\psi)$. Moreover, the resulting GNS productions capture more of the structure found in real characters than either the raw BPL prior (Fig 1, 4b) or the supplemental nonparametric BPL prior [29]. In Fig. 4b we show a grid of 36 new character concepts sampled from our generative model at reduced temperature setting $T = 0.5$ (Appendix A.2). The model produces characters in multiple distinct styles, with some having more angular, line-based structure and others relying on complex curves. In Appendix Fig. A14, we show a larger set of characters sampled from GNS, plotted in a topologically-organized grid alongside a corresponding grid of "nearest neighbor" training examples. In many cases, samples from the model have a distinct style and are visually dissimilar from their nearest Omniglot neighbor.

**Marginal image likelihoods.** As a final evaluation, we computed likelihoods of held-out character images by marginalizing over the latent type and token variables of GNS to estimate $P(I) = \int P(\psi, \theta, I)\partial\psi\partial\theta$. We hypothesized that our causal generative model of character concepts would yield better test likelihoods compared to deep generative models trained directly on image pixels. As detailed in Appendix D, under the minimal assumption that our $K$ posterior parses represent sharply peaked modes of the joint density, we can obtain an approximate lower bound on the marginal $P(I)$ by using Laplace's method to estimate the integral around each mode and summing the resulting integrals. In Table 3, we report average log-likelihood (LL) bounds obtained from GNS for a random subset of 1000 evaluation images, compared against test LL bounds from both the SG [43] and the VHE [21] models. Our GNS model performs stronger than each alternative, reporting the best overall log-likelihood per dimension.

Table 3: Test log-likelihood bounds.

| Model | Im. Size | LL | LL/dim |
|-------|----------|------|--------|
| VHE | 28x28 | -61.2 | -0.0496 |
| SG | 52x52 | -134.1 | -0.0781 |
| GNS | 105x105 | -383.2 | **-0.0348** |

## 6 DISCUSSION

We introduced a new generative neuro-symbolic (GNS) model for learning flexible, task-general representations of character concepts. We demonstrated GNS on the Omniglot challenge, showing that it performs a variety of inductive tasks in ways difficult to distinguish from human behavior.

Some evaluations were still qualitative, and future work will further quantify these results using Visual Turing Tests [29].

Whereas many machine learning algorithms emphasize breadth of data domains, isolating just a single task across datasets, we have focused our efforts in this paper on a single domain, emphasizing depth of the representation learned. Human concept learning is distinguished for having both a breadth and depth of applications [37, 31], and ultimately, we would like to capture both of these unique qualities. We see our character model as belonging to a broader class of generative neuro-symbolic (GNS) models for capturing the data generation process. We have designed our model based on general principles of visual concepts—namely, that concepts are composed of reusable parts and locations—and we describe how it generalizes to 3D object concepts in Appendix E. As in the human mind, machine learning practitioners have far more prior knowledge about some domains vs. others. Handwritten characters is a domain with strong priors [1, 33, 23], implemented directly in the human mind and body. For concepts like these with more explicit causal knowledge, it is beneficial to include priors about how causal generative factors translate into observations, as endowed to our character model through its symbolic rendering engine. For other types of concepts where these processes are less clear, it may be appropriate to use more generic neural networks that generate concepts and parts directly as raw stimuli, using less symbolic machinery and prior knowledge. We anticipate that GNS can flexibly model concepts in both types of domains, although further experiments are needed to demonstrate this.

Our current token model for character concepts is much too simple, and we acknowledge a few important shortcomings. First, as shown in Appendix Fig. A10, there are a number of scenarios in which the parses from a training character cannot adequately refit to a new example of the same character without a token model that allows for changes to discrete variables. By incorporating this allowance in future work, we hope to capture more knowledge in this domain and further improve performance. Furthermore, although our vision for GNS is to represent both concepts and background knowledge with neuro-symbolic components, the current token-level model uses only simple parametric distributions. In future work, we hope to incorporate token-level models that use neural network sub-routines, as in the type-level model presented here.

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

## A    GENERATIVE MODEL

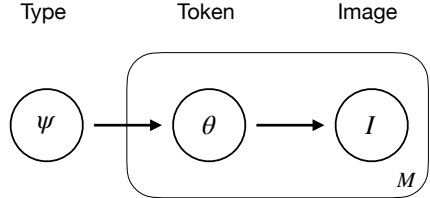

Figure A5: The GNS hierarchical generative model.

The full hierarchical generative model of GNS is depicted in Fig. A5. The joint density for type $\psi$, token $\theta^{(m)}$, and image $I^{(m)}$ factors as

$$P(\psi, \theta^{(m)}, I^{(m)}) = P(\psi)P(\theta^{(m)}|\psi)P(I^{(m)}|\theta^{(m)}).   \qquad (6)$$

The type $\psi$ parameterizes a motor program for generating character tokens $\theta^{(m)}$, unique exemplars of the concept. Both $\psi$ and $\theta^{(m)}$ are expressed as causal drawing parameters. An image $I^{(m)}$ is obtained from token $\theta^{(m)}$ by rendering the drawing parameters and sampling binary pixel values.

### A.1    TRAINING ON CAUSAL DRAWING DATA

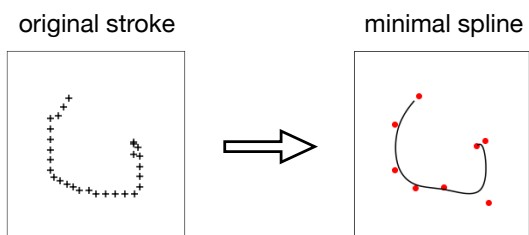

Figure A6: Spline representation. Raw strokes (left) are converted into minimal splines (right) using least-squares optimization. Crosses (left) indicate pen locations and red dots (right) indicate spline control points.

To learn the parameters of $P(\psi)$ and $P(\theta^{(m)} \mid \psi)$, we fit our models to the human drawing data from the Omniglot background set. In this drawing data, a character is represented as a variable-length sequence of strokes, and each stroke is a variable-length sequence of pen locations $\{z_1, ..., z_T\}$, with $z_t \in \mathbb{R}^2$ (Fig. A6, left). Before training our model on background drawings, we convert each stroke into a minimal spline representation using least-squares optimization (Fig. A6, right), borrowing the B-spline tools from [29]. The number of spline control points depends on the stroke complexity and is determined by a residual threshold. Furthermore, we removed small strokes using a threshold on the trajectory length. These processing steps help suppress noise and emphasize signal in the drawings. Our generative models are trained to produce character drawings, where each drawing is represented as an ordered set of splines (strokes). The number of strokes, and the number of spline coordinates per stroke, are allowed to vary in the model.

### A.2    TYPE PRIOR

The type prior $P(\psi)$ represents a character as a sequence of strokes, with each stroke decomposed into a starting location $y_i \in \mathbb{R}^2$, conveying the first spline control point, and a stroke trajectory $x_i = \{\Delta_1, ..., \Delta_N\}$, conveying deltas between spline control points. It generates character types one stroke at a time, using a symbolic rendering procedure called $f_{\mathrm{render}}$ as an intermediate processing step after forming each stroke. An image canvas $C$ is used as a memory state to convey information about previous strokes. At each step $i$, the next stroke's starting location and trajectory are sampled with procedure `GeneratePart`. In this procedure, the current image canvas $C$ is first read by the *location model* (Fig. 2), a convolutional neural network (CNN) that processes the image and returns a

probability distribution for starting location $y_i$:

$$y_i \sim p(y_i \mid C).$$

The starting location $y_i$ is then passed along with the image canvas $C$ to the *stroke model*, a Long Short-Term Memory (LSTM) architecture with a CNN-based image attention mechanism.The stroke model samples the next stroke trajectory $x_i$ sequentially one offset at a time, selectively attending to different parts of the image canvas at each sample step and combining this information with the context of $y_i$:

$$x_i \sim p(x_i \mid y_i, C).$$

After `GeneratePart` returns, the stroke parameters $y_i, x_i$ are rendered to produce an updated canvas $C = f_{\text{render}}(y_i, x_i, C)$. The new canvas is then fed to the *termination model*, a CNN architecture that samples a binary termination indicator $v_i$:

$$v_i \sim p(v_i \mid C).$$

Both our location model and stroke model follow a technique from [16], who proposed to use neural networks with mixture outputs to model handwriting data. Parameters $\{\pi^{1:K}, \mu^{1:K}, \sigma^{1:K}, \rho^{1:K}\}$ output by our network specify a Gaussian mixture model (GMM) with K components (Fig. 2; colored ellipsoids), where $\pi^k \in (0, 1)$ is the mixture weight of the $k^{\text{th}}$ component, $\mu^k \in \mathbb{R}^2$ its means, $\sigma^k \in \mathbb{R}_+^2$ its standard deviations, and $\rho^k \in (-1, 1)$ its correlation. In our location model, a single GMM describes the distribution $p(y_i \mid C)$. In our stroke model, the LSTM outputs one GMM at each timestep, describing $p(\Delta_t \mid \Delta_{1:t-1}, y_i, C)$. The termination model CNN has no mixture outputs; it predicts a single Bernoulli probability to sample binary variable $v_i$. When sampling from the model at test time, we use a temperature parameter proposed by Ha & Eck [20] (see [20, Eq. 8]) to control the entropy of the mixture density outputs.

### A.3 TOKEN MODEL

---

**procedure** GENERATETOKEN($\psi$)
    $\{\kappa, y_{1:\kappa}, x_{1:\kappa}\} \leftarrow \psi$                   ▷ Unpack type-level variables
    **for** $i = 1 \ldots \kappa$ **do**
        $y_i^{(m)} \sim P(y_i^{(m)} \mid y_i)$            ▷ Sample token-level location
        $x_i^{(m)} \sim P(x_i^{(m)} \mid x_i)$              ▷ Sample token-level part
    $A^{(m)} \sim P(A^{(m)})$             ▷ Sample affine warp transformation
    $\theta \leftarrow \{y_{1:\kappa}^{(m)}, x_{1:\kappa}^{(m)}, A^{(m)}\}$
    **return** $\theta$                               ▷ Return concept token

---

Figure A7: Token model sampling procedure.

Character types $\psi$ are used to parameterize the procedure `GenerateToken`($\psi$), a probabilistic program representation of token model $P(\theta^{(m)} \mid \psi)$. The psuedo-code of this sampling procedure is provided in Fig. A7. The location model $P(y_i^{(m)} \mid y_i)$ and part model $P(x_i^{(m)} \mid x_i)$ are each zero-mean Gaussians, with standard deviations fit to the background drawings following the procedure of Lake et al. [29] (see SM 2.3.3). The location model adds noise to the start of each stroke, and the part model adds isotropic noise to the 2d cooridnates of each spline control point in a stroke. In the affine warp $A^{(m)} \in \mathbb{R}^4$, the first two dimensions control global re-scaling of spline coordinates, and the second two control a global translation of the center of mass. The distribution is

$$P(A^{(m)}) = \mathcal{N}([1, 1, 0, 0], \Sigma_A), \tag{7}$$

with the parameter $\Sigma_A$ similarly fit from background drawings (see SM 2.3.4 in [29]).

## B APPROXIMATE POSTERIOR

To obtain parses $\{\psi, \theta\}_{1:K}$ for our approximate posterior (Eq. 3) given an image $I$, we follow the high-level strategy of Lake et al. [29], using fast bottom-up search followed by discrete selection and continuous optimization. The algorithm proceeds by the following steps.

**Step 1:** Propose a range of candidate parses with fast bottom-up methods. The bottom-up algorithm extracts an undirected skeleton graph from the character image and uses random walks on the graph to propose a range of candidate parses. There are typically about 10-100 proposal parses, depending on character complexity (Fig. A8).

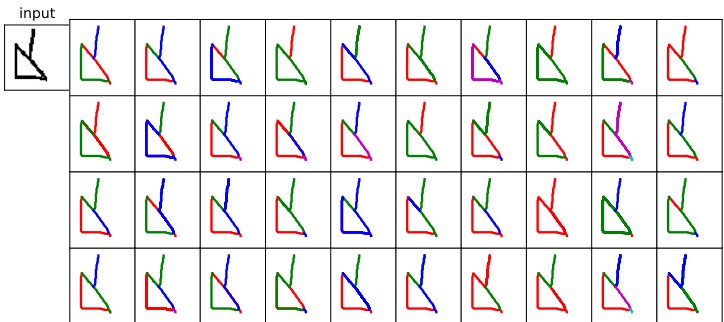

Figure A8: The initial "base" parses proposed for an image with skeleton extraction and random walks.

**Step 2:** Select stroke order and stroke directions for each parse using exhaustive search with the type prior $P(\psi)$. Random search is used for complex parses with large configuration spaces.

**Step 3:** Score each of the proposal parses using type prior $P(\psi)$ and select the top-$K$ parses. We use $K = 5$ following previous work [29].

**Step 4:** Separate each parse into type and token $\{\psi, \theta\}$ and optimize the continuous type- and token-level parameters with gradient descent to maximize the full joint density $P(\psi, \theta, I)$ of Eq. 1.

**Step 5:** Compute weights $\pi_{1:K}$ for each parse by computing $\tilde{\pi}_k = P(\psi_k, \theta_k, I)$ and normalizing $\pi_k = \tilde{\pi}_k / \sum_{k=1}^{K} \tilde{\pi}_k$.

## C  INFERENCE FOR GENERATING NEW EXEMPLARS

When generating new exemplars, we are given a single image $I^{(1)}$ of a novel class and asked to generate new instances $I^{(2)}$ (overloading the parenthesis notation from classification). To perform this task with GNS, we first sample from our approximate posterior $Q(\psi, \theta \mid I^{(1)})$ to obtain parse $\{\psi, \theta\}$ (see Eq. 3), and then re-sample token parameters $\theta$ from our token model $P(\theta^{(2)} \mid \psi)$. Due to high-dimensional images, mass in the approximate posterior often concentrates on the single best parse. To model the diversity seen in different human parses, we apply a temperature parameter to the log of unnormalized parse weights $\log(\tilde{\pi}'_k) = \log(\tilde{\pi}_k)/T$ before normalization, selecting $T = 8$ for our experiments. With updated weights $\pi'_{1:K}$ our sampling distribution is written as

$$P(I^{(2)}, \theta^{(2)} \mid I^{(1)}) \approx \sum_{k=1}^{K} \pi'_k P(I^{(2)} \mid \theta^{(2)}) P(\theta^{(2)} \mid \psi_k). \tag{8}$$

## D  MARGINAL IMAGE LIKELIHOODS

Let $z = \psi \cup \theta$ be a stand-in for the joint set of type- and token-level random variables in our GNS generative model. The latent $z$ includes both continuous and discrete variables: the number of strokes $\kappa$ and the number of control points per stroke $d_{1:\kappa}$ are discrete, and all remaining variables are continuous. Decomposing $z$ into its discrete variables $z_D \in \Omega_D$ and continuous variables $z_C \in \Omega_C$, the marginal density for an image $I$ is written as

$$P(I) = \sum_{z_D \in \Omega_D} \int P(I, z_D, z_C) \partial z_C. \tag{9}$$

For any subset $\tilde{\Omega}_D \subset \Omega_D$ of the discrete domain, the following inequality holds:

$$P(I) \geq \sum_{z_D \in \tilde{\Omega}_D} \int P(I, z_D, z_C) \partial z_C. \tag{10}$$

Our approximate posterior (Eq. 3) gives us $K$ parses that represent promising modes $\{z_D, z_C\}_{1:K}$ of the joint density $P(I, z_D, z_C)$ for an image $I$, and by setting $\tilde{\Omega}_D = \{z_D\}_{1:K}$ to be the set of $K$ unique discrete configurations from our parses, we can compute the lower bound of Eq. 10 by computing the integral $\int P(I, z_D, z_C) \partial z_C$ at each of these $z_D$.

At each $z_{Dk} \in \{z_D\}_{1:K}$, the log-density function $f(z_C) = \log P(I, z_{Dk}, z_C)$ has a gradient-free maximum at $z_{Ck}$, the continuous configuration of the corresponding posterior parse. These maxima were identified by our gradient-based continuous optimizer during parse selection (Appendix B). If we assume that these maxima are sharply peaked, then we can use Laplace's method to estimate the integral $\int P(I, z_{Dk}, z_C) \partial z_C$ at each $z_{Dk}$. Laplace's method uses Taylor expansion to approximate the integral of $e^{f(x)}$ for a twice-differentiable function $f$ around a maximum $x_0$ as

$$\int e^{f(x)} \partial x \approx e^{f(x_0)} \frac{(2\pi)^{\frac{d}{2}}}{|-H_f(x_0)|^{\frac{1}{2}}}, \tag{11}$$

where $x \in \mathbb{R}^d$ and $H_f(x_0)$ is the Hessian matrix of $f$ evaluated at $x_0$. Our log-density function $f(z_C)$ is fully differentiable w.r.t. continuous parameters $z_C$, therefore we can compute $H(z_C) = \partial^2 f / \partial z_C^2$ with ease. Our approximate lower bound on $P(I)$ is therefore written as the sum of Laplace approximations at our $K$ parses:

$$P(I) \geq \sum_{k=1}^{K} \int P(I, z_{Dk}, z_C) \partial z_C \approx \sum_{k=1}^{K} P(I, z_{Dk}, z_{Ck}) \frac{(2\pi)^{\frac{d}{2}}}{|-H(z_{Ck})|^{\frac{1}{2}}} \tag{12}$$

## E  APPLYING GNS TO 3D OBJECT CONCEPTS

The GNS modeling framework is designed to capture inductive biases for concept learning that generalize across different types of visual concepts. We are actively working on applying GNS as a task-general generative model of 3D object concepts, such as chairs and vehicles, again with a neuro-symbolic generative process for parts and relations. Here, we briefly review the path forward for training GNS models of object concepts.

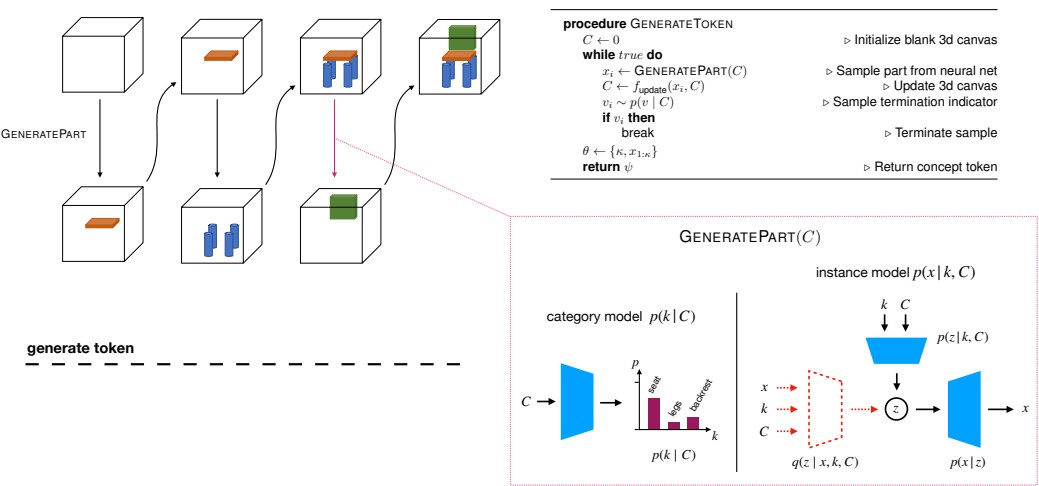

Figure A9: A GNS model for the concept of a chair.

Everyday 3D object concepts have more within-class variability than handwritten characters; for example, different chair *tokens* vary in the number of arm rests, legs, etc. much as different handwritten character *types* vary in their parts, although both domains are characterized by similar structural

and statistical considerations. To address these differences within GNS, we shift up one level in the generative hierarchy and design a token-level model for generating new exemplars of an individual concept (chairs, cars, etc.) that mirrors our type-level model for characters. The architecture and sampling procedure of GNS for 3D object tokens is given in Fig. A9. As with characters, our object model produces samples one part at a time, using a 3D canvas $C$ in place of the previous 2D canvas. The procedure `GeneratePart` consists of two neural network components: 1) a discriminitively-trained *category model* $p(k \mid C)$ that predicts the category label $k$ of the next part given the current canvas (leg, arm, back, etc.), and 2) a generative *instance model* $p(x \mid k, C)$ that is trained with a variational autoencoder objective to sample an instance of the next part $x$ given the current canvas and predicted part category label. Objects and object parts are represented as 3D voxel grids, and all neural modules, including the category model and the encoder/decoder of the instance model, are parameterized by 3D convolutional neural networks. A function $f_{\text{update}}$ is used to update the current canvas with the most recent part by summing the voxel grids. A GNS model for a particular concept is trained on examples of the 3D voxel grids with semantic part labels.

## F    EXPERIMENTS: SUPPLEMENTAL FIGURES

### F.1    ONE-SHOT CLASSIFICATION

In Fig. A10 we show a collection of GNS fits from 7 different classification trials, including 2 trials that were misclassified (a misrepresentative proportion).

### F.2    PARSING

In Fig. A11 we show a collection of predicted parses from GNS for 100 different target images.

### F.3    GENERATING NEW EXEMPLARS

Fig. A12 shows new exemplars produced by GNS for 12 different target images, and Fig. A13 shows new exemplars produced by two alternative neural models from prior work on Omniglot.

### F.4    GENERATING NEW CONCEPTS (UNCONSTRAINED)

In Fig. A14 we show a grid of 100 new character concepts produced by GNS, plotted alongside a corresponding grid of "nearest neighbor" training examples.

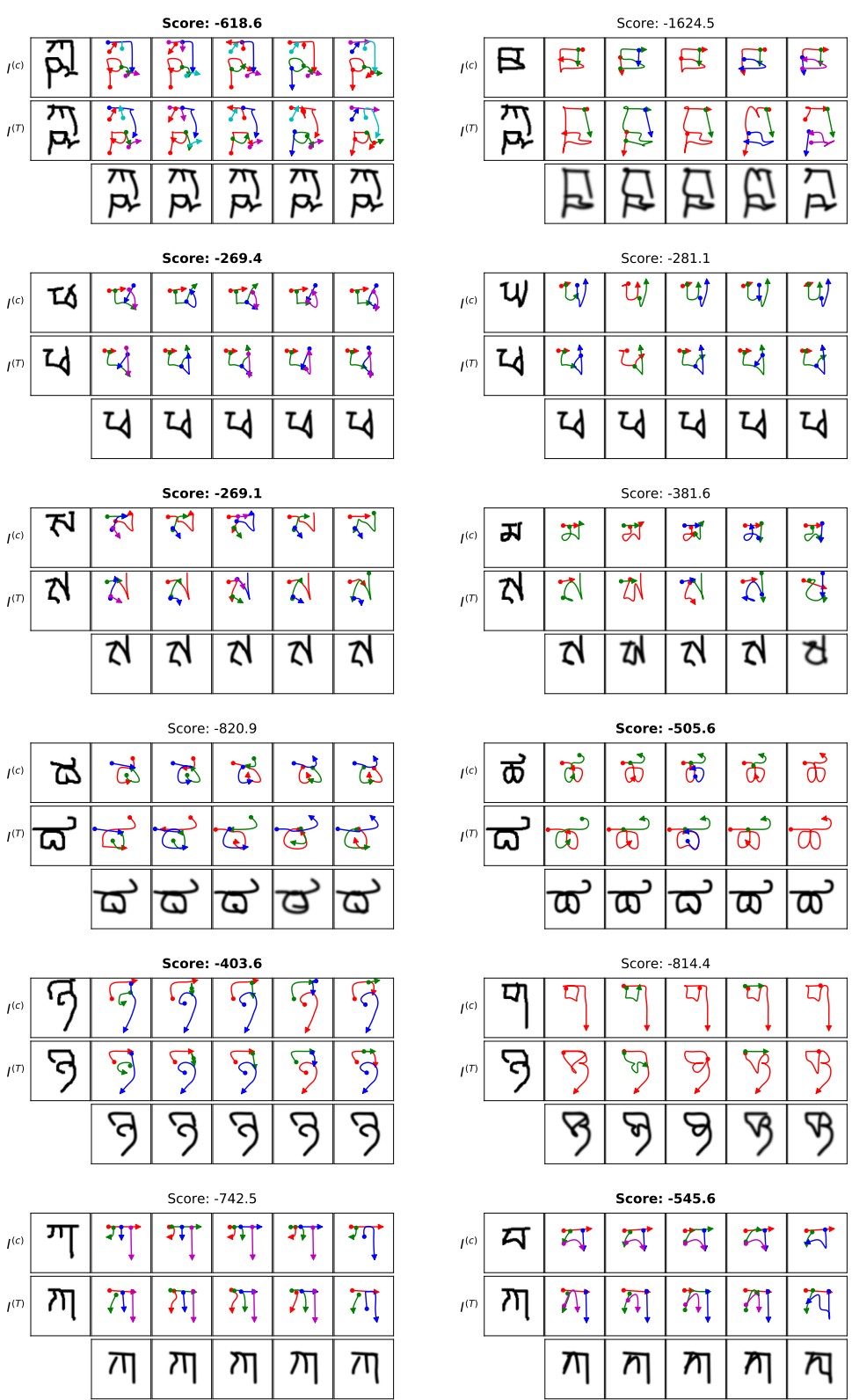

Figure A10: Classification fits. Each row corresponds to one classification trial (one test image). The first column shows parses from the correct training image re-fit to the test example, and the second column parses from an incorrect training image. The two-way score for each train-test pair is shown above the grid, and the model's selected match is emboldened. The 4th and 6th row here are misclassified trials.

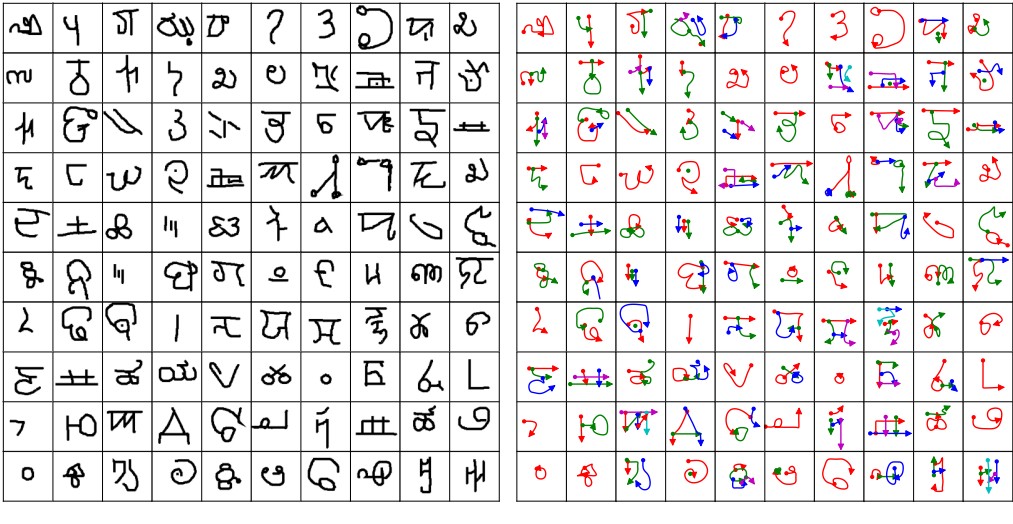

(a) Target images

(b) GNS parses

Figure A11: Parsing. GNS predicted parses for 100 character images selected at random from the Omniglot evaluation set. (a) A 10x10 grid of target images. (b) A corresponding grid of GNS predicted parses per target image.

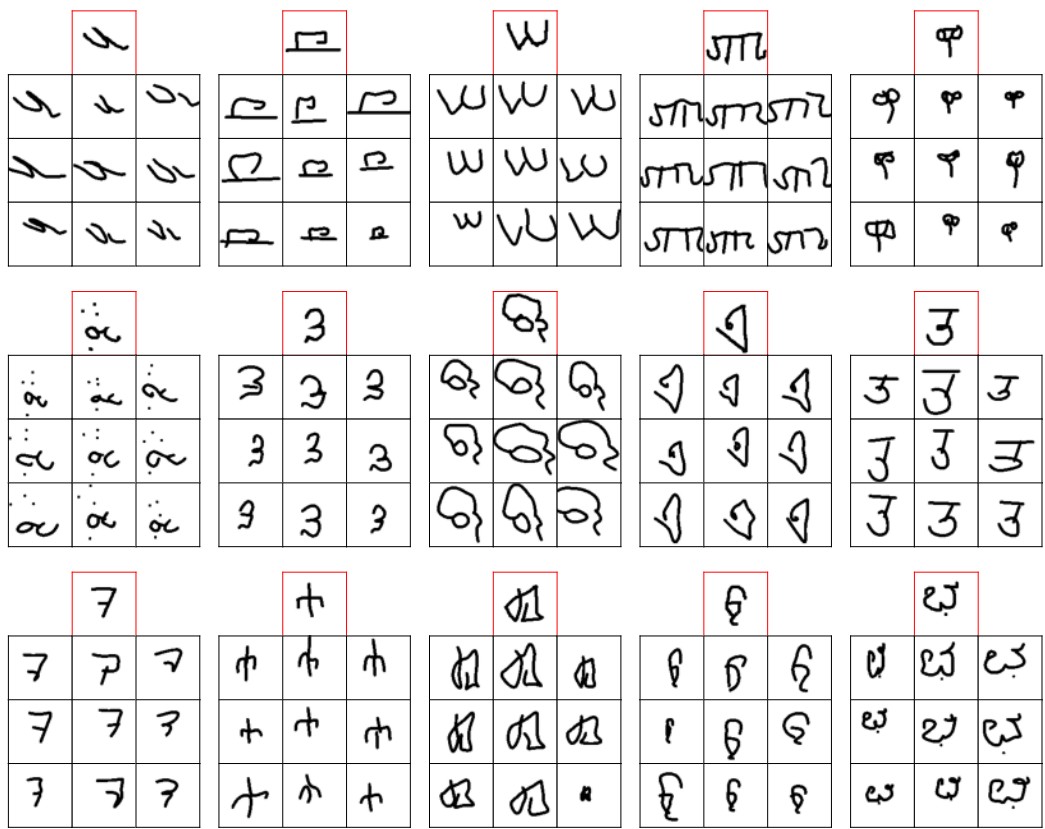

Figure A12: Generating new exemplars with GNS. Twelve target images are highlighted in red boxes. For each target image, the GNS model sampled 9 new exemplars, shown in a 3x3 grid under the target.

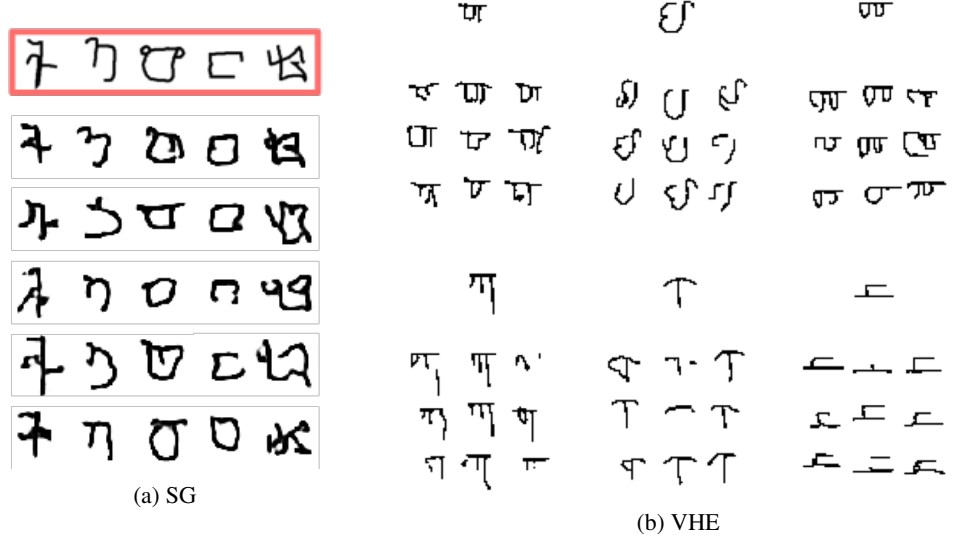

(a) SG

(b) VHE

Figure A13: New exemplars produced by the Sequential Generative (SG) model [43] and the Variational Homoencoder (VHE) [21]. (a) The SG model shows far too much variability, drawing what is clearly the wrong character in many cases (e.g. right-most column). (b) The VHE character samples are often incomplete, missing important strokes of the target class.

GNS samples     Omniglot neighbors

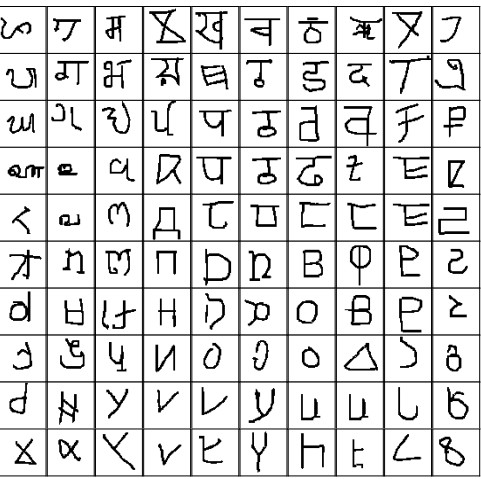

Figure A14: Generating new concepts (unconstrained). 100 new concepts sampled unconditionally from GNS are shown in a topologically-organized grid alongside a corresponding grid of "nearest neighbor" training examples. To identify nearest neighbors, we used cosine distance in the last hidden layer of a CNN classifier as a metric of perceptual similarity. The CNN was trained to classify characters from the Omniglot background set, a 964-way classification task.

