# OpenReview forum: "Learning Task-General Representations with Generative Neuro-Symbolic Modeling"
_ICLR.cc/2021/Conference — ICLR 2021 Poster_

### Official Review · AnonReviewer4 · 2020-10-26
**Interesting approach, but some strange choices, and doesn't quite fulfill its own promises**

**Rating:** 7
**Confidence:** 4

**Review:**

### Summary

In this paper, the authors seek to combine the advantages of symbolic, compositional models and neural approaches, in particular by using "probabilistic programs with neural network subroutines" which should reflect a causal generative process. This is certainly an interesting area to explore. But I'm not convinced that the approach they propose (GNS) lives up to their stated goals.

The concrete model they describe (a generative model of Omniglot characters) seems to be essentially a recurrent neural controller whose outputs are connected to a differentiable stroke-rendering output function, and it seems to me to have very little "symbolic" structure. There is also a hierarchy of "type" and "token" levels (similar to BPL) but this hierarchy makes strong independence assumptions and seems too low-level to capture the true generative process. The authors claim this is a framework for learning task-general representations, but I struggle to see what this framework consists of; their discussion of GNS in section 3 seems highly tailored to the Omniglot task.

The paper is well written and the model itself is described in detail. The experimental results on four Omniglot tasks are strong and quite thorough, and the GNS model does seem to outperform the BPL model and other baselines. The authors also acknowledge some of the shortcomings of their model and describe some interesting possible future extensions.

I lean toward accepting the paper, since it is an interesting approach to the Omniglot tasks and does have some solid contributions. But, at least in it's current state, I feel like the paper falls short of fulfilling its own promises.

### Detailed comments

The introduction and motivation sections are very well written, and put forth a vision of Generative Neuro-Symbolic Modeling (GNS) as a way to combine the advantages of symbolic approaches with learned neural ones. The authors describe how symbolic models can generalize well and behave compositionally, but only when their strong assumptions correctly match the data-generating process; on the other hand, neural models can flexibly learn from data but can struggle to capture compositionality and may not generalize as well. GNS is described as a way to unify these strengths and obtain flexible models that also generalize.

The high-level goals of this approach seem to overlap to some degree with the HOUDINI framework [1], in that both are trying to combine the advantages of neural and symbolic approaches for generalizing to new tasks. There are significant differences, of course, but it is likely worth a brief discussion in the related work section.

The generative model proposed in section 3 seems strange to me in a couple of ways. Like BPL, it is separated into a "type" level, which is meant to capture character concepts, and a "token" level, which is meant to capture individual drawings of each character. In BPL, the "type" level values are structured objects that combine sequences of subparts and link them with discrete relationships, and the "token" level values are collections of strokes that are then rendered. In GNS, however, both the "type" and "token" levels are collections of rendered strokes, and the authors note that the type prior alone already specifies an "auto-regressive density function that can evaluate exact likelihoods of character drawings". The "token" model simply perturbs the strokes from the "type" level with simple 2D Gaussians, independently for each control point in the stroke, and then applies a global affine transformation. The authors claim that this is a causal model of character drawings, but it doesn't match my intuition about what the causal generative process of those drawings should be. In particular, it seems to me that the character concept needs to include the kinds of relational structure captured by BPL, at least implicitly. The representation used by GNS seems too low-level to adequately capture a character concept, and the strong assumption of independent perturbations leads to the loss of certain structural elements (as the authors note in section 5 under "Generating new exemplars").

Given the strong claims made initially, I was really hoping for a more powerful neural token model combined with a more abstract type model. Perhaps this could look something like Sketch-RNN [2] or DRAW, with a "type" prior over an abstract latent space and then a "token" decoder that renders strokes (similar to the proposed GNS type model). It seems to me that that would still having many of the same advantages but would also have the ability to generate more flexible output “tokens” than the GNS model (for instance, it might learn to connect strokes to previous strokes even when those previous strokes are perturbed by noise). The VAE framework might also provide a nice way of generalizing the very-domain-specific bottom-up parse proposal distribution described in the appendix.

More broadly, I get the sense from the introduction that this generative model of sketches is intended to be an instance of a more general framework of some kind (although this isn't particularly obvious, since the sketch-specific model is frequently referred to as simply "GNS" in the paper). But it is not at all clear to me what this framework actually is. On the most general side, is any model that composes neural submodules according to domain knowledge an instance of the framework? This is the impression that I got from the introduction, but it would seem to include a large number of preexisting models; for instance, the DRAW model specifies a structured probabilistic generation process of images. Or perhaps it requires combining neural and domain-specific primitives together into differentiable processes? This would include various papers on differentiable rendering (such as [3] and [4]) or audio synthesis (such as [5]). On the other hand, the conclusion suggests that the GNS framework is only applicable for visual tasks, which is a considerably smaller scope than the introduction suggested. (The authors also include some brief discussion of a GNS model of 3d structures in the appendix, but it seems very different from the model for characters, and I'm not sure what the common thread is.) The paper could be much improved by being more explicit about what the claimed general concept-learning framework actually consists of.

### Questions and suggestions

I was a bit surprised that neural architectures are referred to as "nonparametric". I suppose this is because vastly overparameterized networks behave like nonparametric models, despite having parameters? It might be worth clarifying the meaning here to avoid confusion.

Is equation (4) doing maximum a-posteriori (MAP) inference over the token-level parameters? It would be useful to go into a bit more detail on what this approximation is and why approximating in this way is reasonable.

In section 5 under "Parsing", it wasn't clear to me why having a variety of parses is bad for the first character but desirable for the second one. Is it because that matches what humans do?

Minor issue: I noticed that the authors appear to have used the ICLR 2020 style files instead of the ICLR 2021 style files.

### References

[1] Valkov, Lazar, et al. "Houdini: Lifelong learning as program synthesis." Advances in Neural Information Processing Systems. 2018.

[2] Ha, David, and Douglas Eck. "A neural representation of sketch drawings." arXiv preprint arXiv:1704.03477 (2017).

[3] Li, Tzu-Mao, et al. "Differentiable monte carlo ray tracing through edge sampling." ACM Transactions on Graphics (TOG) 37.6 (2018): 1-11.

[4] Thies, Justus, Michael Zollhöfer, and Matthias Nießner. "Deferred neural rendering: Image synthesis using neural textures." ACM Transactions on Graphics (TOG) 38.4 (2019): 1-12.

[5] Engel, Jesse, et al. "DDSP: Differentiable digital signal processing." arXiv preprint arXiv:2001.04643 (2020).

----
After reading the author responses and updated manuscript, I have raised my score from 6 to 7.

---

> ### Author Response · Authors · 2020-11-13
> **Clarifications and proposed revisions: R4**
>
> Thank you for the thorough and thoughtful review. We recognize that the current work does not yet realize the full promise of the vision in the introduction, and we’ve suggested revising our paper introduction to reflect the scope of our experiments (main review, point #1). We have also clarified the symbolic structure of our model (point #2) and suggested to revise the paper with these clarifications. We feel that there is a misunderstanding about the meaning of “task-general” and suggest a clarification below. Would these revisions improve our paper in your eyes?
>
> We recognize that our current work is limited to handwritten characters. We did not intend the term “task-general” to refer to a representation that generalizes across different conceptual domains, such as from handwritten characters to vehicles or animals. Research suggests that the structure and inductive biases of human conceptual representations varies greatly across different concept families, and we believe that those of a computational model should as well. We use the term “task-general” to refer to a representation of an individual concept that supports a variety of tasks such as recognition, parsing/segmentation, and generation. Our motivation is to provide a computational account for how human learners quickly grasp new concepts and use them in a variety of tasks, an ability that has eluded prior models from machine learning.
>
> We agree that our generative model of characters may not accurately reflect the causal generative process of human handwriting with complete fidelity, and we discuss some of the current shortcomings in our paper, especially those of the token-level model. In future work, we hope to update the token model to capture important relational structure as you suggest (and we discuss). Nevertheless, the current version of our model far outperforms alternative deep generative models for human-like one-shot generation, and it possesses critical improvements over BPL (main review point #3).
>
> We see Sketch-RNN and DRAW as important prior art, but as having critical shortcomings that we aim to overcome. We discuss Sketch-RNN in our response to R1, and we discuss DRAW and its updated variant Sequential Generative (SG) model throughout our paper. The minimal priors expressed through standard normals in VAEs have thus far been insufficient to account for the structured nature of human background knowledge and inductive biases. Moreover, while models like DRAW and SG have minimal symbolic structures endowed through a notion of “location,” there is no representation of the parts (strokes) in a character, and this leads them to generate new examples that are inconsistent in many cases (see Fig. A13(a)). The lack of compositional representation is also evident when looking at the sample generation process of DRAW/SG, which can be viewed real-time in the authors’ videos. The drawing process is highly sporadic and bears little resemblance to human character generation.
>
> Replies to questions/suggestions:
> - RE nonparametric: Training neural networks to estimate arbitrary density functions and decision surfaces has long been considered an example of nonparametric or “model-free” statistical inference. See Geman et al. [1].
>
> - RE equation 4: Yes, we are doing MAP estimation of $\theta^{(T)}$ at all possible values of the latent $\psi$ (weighted by their posterior probability). In practice we only consider K=5 values of $\psi$, since our posterior approximation is discrete. The classification rule is a modified variant of the one proposed in [2], who provide a complete derivation on pg. 209. We will clarify the details of the approximation in our revision.
>
> - RE parsing: The first character is very simple and has a clear front-runner for stroke composition - this is evident from the human data, which shows that all subjects presented with this character drew it the same way. The second character is a bit more complex, and it was drawn in different ways by different human subjects. The GNS model, which is trained on drawings from different human subjects with different styles/preferences, captures uncertainty for this character by predicting a variety of different parses.
>
> - RE style files: Thanks very much for pointing this out! We were unaware of this and will be sure to update the files.
>
> [1] S. Geman, E. Bienenstock, and R. Doursat. Neural networks and the bias/variance dilemma. Neural Computation, 4(1):1-58, 1992.
>
> [2] B.M. Lake. Towards more human-like concept learning in machines: Compositionality, causality, and learning-to-learn. MIT, Cambridge, MA, 2014.

---

> > ### Comment · AnonReviewer4 · 2020-11-16
> > **Response to clarifications**
> >
> > Yes, I think that revising the introduction to more clearly specify the scope of the contribution would be a definite improvement.
> >
> > As for the "symbolic" structure, perhaps this is just quibbling over semantics, but simply having an output representation based on pen strokes doesn't seem inherently symbolic to me. I agree that such a representation is an improvement over pixel-based approaches. But personally I would refer to that kind of thing as "a domain-specific output representation" or perhaps "an approximate, differentiable world model", since it doesn't seem to involve any manipulation of symbols. (Admittedly this is a minor issue and I don't feel that strongly about it.)
> >
> > I understood what you meant by "task-general" (sorry for unclear wording on my part). My comment *"The authors claim this is a framework for learning task-general representations, but I struggle to see what this framework consists of; their discussion of GNS in section 3 seems highly tailored to the Omniglot task"* was directed at the use of the term "framework", since a framework should ideally generalize across domains even if any individual model/representation does not. (I expanded on this comment in the rest of the review, and you have responded to it in your main response.)
> >
> > It seems we agree that the Omniglot model is not exactly a causal model of how humans write (but that it is perhaps a step closer than previous models of Omniglot). Given this, I'd suggest revising the multiple places throughout the paper where you imply that it *is* an accurate causal model, e.g. "devising a GNS model that learns real compositional and causal structure from a background set of human-drawn characters" on page 2.
> >
> > In terms of the overall framing, would it be accurate to say that your (broad) hypothesis is that human concept-learning is based on building approximate generative models of causal processes, and then performing inference in those models? And that a GNS model of, say, Omniglot, is also an approximate generative model of causal processes, which is why it does well at multiple Omniglot tasks? (And by extension, the failure cases of your specific model are tied to the ways in which the causal structure you have imposed is incorrect?) If that is what you are trying to say, maybe it's less important that GNS models have both probabilistic programs and neural components in them, and more important that a GNS generative model can match the "true" causal process as closely as possible (partially by learning, partially by design); in a sense the probabilistic programs are just a convenient way of hand-engineering the structure that human brains encode in some other way (and perhaps have already learned from "pre-training" on other tasks like drawing pictures).

---

> > > ### Author Response · Authors · 2020-11-23
> > > **Updated manuscript**
> > >
> > > Thanks for the quick and thoughtful reply. We have submitted a revised manuscript and would like to know if our changes have addressed your concerns. Please see our general comment for a summary of the revisions (and excerpts).
> > >
> > > We agree with your comment about the language of “real” compositional/causal structure: we did not intend to convey that our model reflects the human generative process with complete fidelity, and we’ve revised the wording. Our point here is more about the goal of our model (rather than the result), which is fundamentally different from those of other Omniglot models. Other generative models applied to Omniglot have ignored the human drawing data entirely, instead forming their own internal hypotheses about the generative process that produces character images. We take the distinct approach of training on causal drawing data (Appendix A.1), with the goal of building a model of how the data are actually generated.
> > >
> > > P.S. – we added a citation to HOUDINI in our related work, which falls under the category of “input-output” program learning. Thank you for the suggestion.

---

> > > > ### Comment · AnonReviewer4 · 2020-11-23
> > > > **Comments on updated manuscript**
> > > >
> > > > Thank you for updating the manuscript. I think the new abstract, introduction, and discussion section are much clearer about the scope of the contribution and about the sense in which the model is symbolic. I have raised my score from 6 to 7.
> > > >
> > > > Two more minor suggestions:
> > > >
> > > > - The last sentence of Figure 2 caption may be missing a word: "Unique exemplars are produced from a character type by sampling from the token model conditioned on ψ, adding motor noise to the drawing parameters and a random affine transformation." Should that be "and *performing* a random affine transformation", or something similar?
> > > > - In section 5 under "Parsing", I would suggest adding to the paper some of the details from your reply above (specifically, emphasizing that the second character was drawn in different ways by human subjects, and that is why GNS's multiple possible parses are desirable for that character.)

---

> > > > > ### Author Response · Authors · 2020-11-24
> > > > > **Additional changes made**
> > > > >
> > > > > Thanks for catching that in Figure 2, and for the suggestion about our Parsing section. We agree and we've updated the manuscript accordingly. Here's an excerpt from the new description of our parsing results:
> > > > >
> > > > > "Compared to BPL, GNS parses possess a few unique desirable qualities. The first character class has an obvious segmentation to the human eye—evidenced by the consistency of human parses in all examples—and the GNS model replicates this consistency across all 9 predicted parses. In contrast, BPL predicts seemingly-unlikely parses for 2 of the examples shown. The second character is more complex, and it was drawn in different styles by different human subjects. The GNS model, which is trained on data from subjects with different styles, captures the uncertainty in this character by predicting a variety of unique parses. BPL, on the other hand, produces a single, ubiquitous segmentation across all 9 examples."

---

### Official Review · AnonReviewer3 · 2020-10-27
**Plausible model of human visual conceptualization**

**Rating:** 7
**Confidence:** 3

**Review:**

Motivated by few-shot learning challenges such as Omniglot, the authors propose a human-like model for learning how to draw visual concepts that consists of three components: (1) a location model for picking the starting point of the next stroke given the current "canvas", (2) a stroke model that continues an existing stroke, (3) a termination model that decides when to stop drawing.  The three components of this model are trained on example glyphs that are heuristically parsed into strokes.  The authors evaluate performance of this method, called GNS (Generative Neuro-Symbolic) versus more classical program learning approaches based on pre-existing libraries of subroutines and more generic deep neural network architectures.  This approach (1) represents an advance over methods based on pre-defined subroutines in the sense that primitives such as "draw a stroke intersecting an existing stroke" are learned rather than supplied by the programmer and (2) performs much better at few-shot learning of glyphs than generic approaches.

Quality: The idea behind the method makes sense.  The authors evaluate the model on four different tasks, and compare against reasonable baselines.  They also generalize beyond glyphs to learning 3D objects, although results are not shown.  The authors are careful to describe details such preprocessing of the input into splines and candidate parses, although details of the actual architecture do not seem to be given.  The weakest part of the approach is the image model.  It seems to work well enough for performance purposes, but the image distribution involving randomly transforming parts does not seem intuitively plausible as a model of human-like image generation.  As the authors note, this results in their approach underperforming against BPL.  The approach would probably work better if the authors had included stroke variability as a part of the core model.

Clarity:  I had to read the paper a few times to understand the approach.  It would have helped if the authors could better explain the architecture of the various model components.  Besides this issue of vagueness on technical details, the paper is well-written.

Originality:  To my knowledge the use of a separate location and LSTM stroke model for glyphs is novel.  The previous work section is adequate for contextualizing the paper.

Significance:  The paper represents an important approach to mimicking human concept learning.  (I base this on my own intuition; the authors could perhaps make a stronger case for this by including relevant references from the cognitive science literature.)
Chaining together multiple different neural networks to create a new type of ML algorithm has been a promising source of innovation in deep learning, e.g. AlphaGo combining a policy and evaluation network, GANs combining a generator and discriminator.  This paper is yet another example via combining a location and stroke model for learning glyphs.  On the other hand, the authors may have oversold the significance of the work by claiming that this model represents an example of fundamentally new approach in harmonizing artificial neural networks and symbolic program learning.  I am not convinced that this combination of location+stroke model is doing hierarchical concept learning in the same way that humans do.  If this is not what the authors are claiming, they should clarify this in the main paper.

Pros:
 * novel and plausible model for human-like learning
 * achieves reasonable performance on a variety of one-shot learning tasks
 * qualitatively exhibits human-like performance at free glyph generation (in my opinion)

Cons:
 * the type -> token part of the model seems underdeveloped
 * paper is vague on the architectures used for each of the model components (location, stroke, termination)
 * paper makes a claim that the model is doing symbolic modeling, but the term "symbolic modeling" is not well-defined and there is insufficient analysis showing that the model learns something analogous to existing methods such as BPL

---

### Official Review · AnonReviewer1 · 2020-10-29
**Interesting method but the novelty is not enough**

**Rating:** 6
**Confidence:** 4

**Review:**

Summary: This paper presents a generative neuro-symbolic model for learning the task-general representations. The model is inspired by the BPL approach, with less manually-defined prior or rules but more learning-based ingredients. I generally appreciate the ideas and improvements it made. However, compared with BPL, the claims of task-general representation, and neural-symbolic modeling does not match the actual contributions of the work.

Pros
1. The paper is well-written with its motivations,  methods, and corresponding experimental results. It shows a promising direction of learning a representation and model for multiple tasks, \ie, the Omniglot challenge.

2. The formulation of the proposed GNS and the corresponding method of inference provides a well-established and holistic solution for the visual simple concepts learning and inference.

Cons
1. The claimants of human-level visual understanding, and task-general representations are similar to the BPL methods, which is well-known enough. I think the author should put more effort into comparing the BPL and demonstrates the actual improvement compared with it.

2. The current framework is still under the Bayesian statistical modeling, I don't think it provides a task-general representation since it cannot be transferred to more complex domains and perceptual tasks. Moreover, the symbolic representations and symbolic module in the proposed method are quite trivial. The proposed method does not provide deeper insights into the neural symbolic representations.

3. The causal structures in the model are captures in an autoregressive way, which is similar to lots of existing modules.  I don't see any actual causal problems are solved in the framework.

My main concern is that it put too much space in telling the stories that BPL has already proposed and demonstrated, but not the detailed comparisons with BPL and how the technical contributions and novelty of this specific model can benefit the overall human-level understanding of visual concepts. I would like to raise my scores if the concern is properly addressed.

I raised my scores from 4 to 6 after the author updated their draft and answered my questions. I appreciate their efforts in addressing my concerns and improve the paper. The current version is good enough to be accepted and it also compares with other approaches thoroughly. However, I still feel the symbolic module is too simple in this work and does not distinguish it from other works.

---

> ### Author Response · Authors · 2020-11-13
> **Clarifications and proposed revisions: R1**
>
> Thank you for the thoughtful comments. We feel strongly that our GNS model represents a novel and important contribution vs. BPL, and we are suggesting adding a few clarifications about the distinguishing characteristics and features, as well as some additional comparisons to BPL (see main response point #3). We have also suggested clarifications regarding the symbolic and causal structures of our model (see main response point #2 and response below), and we’ve proposed to modify our introduction to clarify that our experiments include only one conceptual domain (point #1). Do you feel these proposed revisions could satisfy your concerns?
>
> We feel that there is a misunderstanding about the meaning of “task-general representation.”
> It sounds that your interpretation is a representation that transfers to different conceptual domains, e.g. from characters to vehicles or animals. In contrast, we use the term to refer to a representation of a specific concept or conceptual family that generalizes to different ways of using the concept (classification, parsing, generating new exemplars, etc.). The structure and inductive biases of human representations varies greatly across different conceptual domains; arguably, those of machines aiming to mimic them should as well. With GNS, we are interested in modeling this structure directly as opposed to building generic, domain-general learning algorithms. This structure is necessary to account for the flexibility of human concepts in a domain, and we will further clarify this motivation in our revision.
>
> Other works (e.g. [1], [2]) have used autoregressive models like ours with similar stroke primitives to model the causal generative processes of handwriting, and we wouldn’t want to give the impression that this is the main novelty of our work. We will add a paragraph to our related work discussing these works and our distinguishing characteristics, which we outline here. First, our novel architecture for generating handwriting—which uses splines rather than raw strokes, intermediate symbolic rendering, and image-based attention—outperforms alternative models in hold-out likelihoods; this was demonstrated in our supplemental manuscript, and we will include it in the main paper if reviewers would like. Second and most importantly, these alternative models of handwriting have not made a connection to the image; therefore while they can generate handwriting as symbolic coordinates, they cannot explain how people use their causal knowledge to learn new characters from visual presentations, how they infer the strokes of a character seen on paper, or how they generate a new example of an observed character. By combining a powerful autoregressive generative model of handwriting with an image rendering model and algorithms for probabilistic inference, we are able to perform all of these various concept learning tasks.
>
> [1] Graves, A. (2013). Generating sequences with recurrent neural networks. arXiv preprint arXiv:1308.0850.
>
> [2] Ha, D. & Eck, D. (2018). A neural representation of sketch drawings. ICLR.

---

> ### Author Response · Authors · 2020-11-23
> **Please let us know whether your concerns have been addressed in the revised manuscript**
>
> Thanks again for your review. We were encouraged to hear that you might consider raising your score, and we’d appreciate to know whether our revised manuscript and our clarifications have satisfied your concerns. We feel strongly about our contribution relative to BPL; we’ve emphasized the distinguishing characteristics in our revision (see general comment above for excerpts) and added new figures with direct comparisons to BPL in the concept learning tasks. There are now BPL comparisons in every concept learning task (Table 2, Fig. 3, Fig. 4a & 4b).
>
> We’ve also added a discussion of prior autoregressive generative models in our related work section, emphasizing the distinguishing characteristics of the current work:
>
> (Related Work)
> “Other works (e.g. [15,19]) have used autoregressive models like ours with similar stroke primitives to model the causal generative processes of handwriting. We develop a novel architecture for generating handwriting, which represents explicit compositional structure by modeling parts and relations with separate modules and applying intermediate symbolic rendering. Most importantly, these prior models have not made a connection to the image; therefore while they can generate handwriting as symbolic coordinates, they cannot explain how people use their causal knowledge to learn new characters from visual presentations, how they infer the strokes of a character seen on paper, or how they generate a new example of an observed character. By combining a powerful autoregressive model of handwriting with an image likelihood model and algorithms for probabilistic inference, we are able to replicate a spectrum of unique human concept learning behaviors.”

---

### Official Review · AnonReviewer2 · 2020-10-30
**GNS review**

**Rating:** 6
**Confidence:** 4

**Review:**

This paper introduces generative neuro-symbolic modelling, advertised as a probabilistic programming framework in which the distributions are modelled by neural networks. This is a very exciting idea, and well past its time. However, the work discussed here is limited to modelling the drawn characters in the Omniglot dataset, rather than a general framework. It is clear there is not a usable tool that would allow for practical construction of a wide range of probabilistic programs, from the fact that the paper sticks with a simple model for the Omniglot problem, rather than doing a lot of manipulations on the model, and, more importantly, the fact that the inference is specific to the Omniglot model, and is presumably not very fast, given the description. Thus, while the sales pitch to the paper is very exciting, compared to that, the content itself, which is really just a model of one problem, is a bit disappointing. That was also a property of the earlier Lake et al paper - I never understood what the "framework" was there either. That is not to dismiss this paper. The insight for this problem is that, rather than relying on a database of character primitives, like the previous Bayesian model of the problem, you can sequentially generate characters (types)  non-independently using the expected cast of characters (convolutions and LSTMs - although there are also splines involved, so it's not exactly typical CNN either). The evaluations are not all that impressive, except for the unconditional character generation shown at the beginning, which is clearly outstripping the BPL model in terms of human-like ness. In Table 3, I don't understand why different image sizes are used, and why BPL is not compared. The paper is very clear.

In short, I really like the idea in this paper, and I think it is worth building on. The section in the appendix detailing the future work on 3D object modelling sounds very promising. However, the result itself is not all that significant beyond the idea; the evaluations are not particularly strong, except for the generation of novel characters, which would benefit from a formal evaluation, and perhaps a transfer to a less artificial problem.

---

> ### Author Response · Authors · 2020-11-13
> **Clarifications and proposed revisions: R2**
>
> Thank you for your thoughtful review. We understand your concern about the “framework” presentation, and we are suggesting updating the language of our intro to reflect the scope of the current work (see main response, point #1). We have also suggested additional comparisons to BPL (main response, point #3). If these revisions are implemented as described, do you feel they would satisfy your concerns?
>
> We’re happy to hear that you like the idea of probabilistic program representations with neural network distributions. We feel strongly that the material of our paper represents an important first demonstration of this paradigm. Unlike other works in machine learning that aim for general learning algorithms that transfer across domains and datasets with little or no modification, the GNS formula calls for engineering minimal symbolic structures to capture important inductive biases in a new domain. We see these structures as a critical to understanding and reproducing the ways that people quickly grasp new concepts and use them in a variety of tasks, and we are motivated by the shortcomings of works that have taken the alternative approach. A consequence, however, is that it’s a challenge to fit model details and results from multiple domains into a single conference paper.
>
> RE marginal image likelihoods: other models (e.g. SG, VHE) use downsized variants of the Omniglot images to make the learning task easier. Our model uses the original image size of 105x105, so we had to compare results at different sizes. We use the standard "bits per dimension" metric for comparisons of this kind where the likelihood models are from the same parametric family (Bernoulli pixel probabilities) but the number of dimensions varies. The BPL model cannot compute marginal image likelihoods.
>
> RE character evaluations: we evaluated the model on 4 unique conceptual tasks (plus image likelihoods) and provided additional figures in the appendix with extra model samples, parses, and classification fits. We will include additional comparisons to the BPL generative model where possible (see suggestions from main response). We note that the BPL model cannot compute marginal likelihoods (neither for an image or a latent drawing).
>
>
> P.S. - RE the BPL "framework": although Lake et al. (2015) provided only results for characters, BPL has since been demonstrated in other domains, such as recursive visual concepts (Lake & Piantadosi, 2020). Supporting our concern about paper length and model details, each of these applications was communicated in a full-length journal paper.
>
> Lake, B.M. and Piantadosi, S.T. (2020). People infer recursive visual concepts from just a few examples. Computational Brain & Behavior, 3(1), 54-65.

---

### Author Response · Authors · 2020-11-13
**Clarifications and proposal for new introduction**

We thank each of the reviewers for their thoughtful comments. Here, we summarize several common critiques, along with our plans for addressing them in a revised manuscript. While revising the manuscript, we welcome your thoughts on the planned revisions, and whether or not they address your principal concerns. We will then update the paper and provide it for your further review.

1 - Framework motivation

Multiple reviewers (R1, R2, R4) felt that introducing GNS as a "framework" may be overstating the contribution of the current paper. Your points are well-taken, and in response we are prepared to tighten and tone back the broader framing. In the revised introduction, rather than using the word “framework”, we plan to introduce the generative model more directly as a novel approach to tackling the Omniglot Challenge. We will also mention how we see this model as belonging to broader class of Generative Neural-Symbolic (GNS) models that seek to capture the data-generating process through probabilistic programs with neural network sub-routines. GNS is further characterized by a set of principles for building richer, more human-like generative models, by capturing causality, compositionality, and complex correlations in how new exemplars are generated. (A proposed application to 3D object concepts is outlined in the appendix). We recognize that our vision is ambitious and that we are still in the early stages of realizing its full potential; nevertheless, we see our current work as an important step forward. We hope that you agree, and a revised statement of our contributions will mitigate some of your concerns. We welcome your further thoughts on the framing, and we are happy to consider any additional suggestions you may have.

2 - Symbolic structure & causality

Two reviewers had questions related to the symbolic component of the model, with reactions that the symbolic structure was too minimal (R3,R4) or unclear (R3). We are prepared to make revisions that clarify the symbolic structure of our model, its significance and its novelty over existing methods. As a model of character images, our GNS prototype relies on symbolic notions of pen actions (splines, stroke breaks, etc.) and a symbolic rendering engine to capture the causal structure in characters and how observations are produced by the underlying generative procedures. This machinery provides clear advantages over alternative models that operate directly on image pixels, which we demonstrate. Given the failures of existing models without these components, we feel that this machinery is critical to the success of our efforts. For more discussion of the symbolic components and how they relate to prior work, see direct response to R1.

3 - Novelty & improvements over BPL

Reviewers raised concerns about the novelty of our model in relation to prior work on BPL (R1), and the need for additional comparisons (R1, R2). We feel strongly that both the motivation for GNS and the implementation of our character prototype are distinct from BPL, and we’re prepared to emphasize the distinguishing characteristics and include additional comparisons in a revision. Our motivation for GNS is to provide a synthesis of two long-standing traditions in cognitive modeling and AI—symbolic and neural network models—in order to engineer more human-like conceptual capabilities. BPL settles for a fully-symbolic generative model, and it's account of human conceptual priors has clear and critical shortcomings.

Our generative model borrows the type-token hierarchy from BPL, which we see as an important scaffolding for a spectrum of conceptual models. That said, the implementation details of each level in our model differ critically. Our type model is a highly expressive generative model that can capture complex correlation structure in character drawings, and this structure is essential to generating new character concepts in realistic, human-like ways. Moreover, we designed our generative model to be fully differentiable at all levels, enabling powerful new optimization techniques for inference using the gradient of model likelihoods; a demonstration is Fig. A10, where our gradient-based optimization finds promising re-fits for classification. The gradient also enables us to compute marginal image likelihoods, an ability BPL lacks.

We agree that our paper could benefit from more comparisons to BPL, and we propose a few here. Quantitative comparisons are limited given that BPL cannot estimate marginal likelihoods of an image or a latent drawing. In our current work, we've included comparisons to BPL for one-shot classification (Table 2) and parsing (Figure 3b). In our revision, we are suggesting including samples from BPL for the "generating new exemplars" task plotted next to our own in Figure 4(a), and additional BPL samples for the "generating new concepts" task--some of which are shown in Figure 1--next to our own in either Figure 4(b) or the appendix, space permitting.

---

### Author Response · Authors · 2020-11-23
**Please see revised manuscript**

Thank you again for the thoughtful and thorough reviews. We’ve made changes to the manuscript based your feedback, and we’d appreciate it if you can take a look at the revisions and let us know if your concerns have been adequately addressed.

===============

We’ve updated the abstract, introduction and discussion sections to reflect a focus on characters in the current work, removing the term “framework.” In doing so we’ve also emphasized and clarified the symbolic structure of our model. Some examples of the revision are as follows:

(Abstract)
“… We bring together these two traditions to learn generative models of concepts that capture rich compositional and causal structure, while learning from raw data. We develop a generative neuro-symbolic (GNS) model of handwritten character concepts that uses the control flow of a probabilistic program, coupled with symbolic stroke primitives and a symbolic image renderer, to represent the causal and compositional processes by which characters are formed. The distributions of parts (strokes), and correlations between parts, are modeled with neural network subroutines, allowing the model to learn directly from raw data and express nonparametric statistical relationships. We apply our model to the Omniglot challenge…”

(Introduction)
“In this paper, we introduce a new approach that leverages the strengths of both the symbolic and neural network paradigms by representing concepts as probabilistic programs with neural network subroutines. We describe an instance of this approach developed for the _Omniglot challenge_ [28] of task-general representation learning and discuss how we see our Omniglot model fitting into a broader class of Generative Neuro-Symbolic (GNS) models that seek to capture the data-generation process. …”

================

We’ve clarified the distinguishing characteristics vs. BPL, re-writing the intro of section 3 (Generative Model) and adding new comparisons to BPL concept learning in Figures 4(a) and 4(b). See the following excerpt from section 3:

(Generative Model)
“Although sharing a common hierarchy, the implementation details of each level in our GNS model differ from BPL in critical ways. The GNS type prior $P(\psi)$ is a highly expressive generative model that uses an external image canvas, coupled with a symbolic rendering engine and an attentive recurrent neural network, to condition future parts on previous and model sophisticated causal and correlational structure. This structure is essential to generating new character concepts in realistic, human-like ways (Sec. 5). Moreover, whereas the BPL model is provided symbolic relations for strokes such as "attach start" and "attach along," GNS learns implicit relational structure from the data, identifying salient patterns in the co-occurrences of parts and locations. Importantly, the GNS generative model is designed to be differentiable at all levels, yielding log-likelihood gradients that enable powerful new inference algorithms (Sec. 4) and estimates of marginal image likelihood (Sec. 5).”

---

### Author Response · Authors · 2020-11-24
**Remaining questions/concerns?**

If you have any additional questions or concerns in light of the revisions, we would love to hear from you while we still have the chance to revise the paper. Thanks for your further consideration of our work.

---

### Decision · Program_Chairs · 2021-01-07
**Final Decision**

**Decision:**

Accept (Poster)

**Comment:**

This paper was reviewed by four experts in the field. Based on the reviewers' feedback, the decision is to recommend the paper for acceptance to ICLR 2021. The reviewers did raise some valuable concerns that should be addressed in the final camera-ready version of the paper. The authors are encouraged to make the necessary changes and include the missing references.